# Methylsulfinyl Hexyl Isothiocyanate (6-MSITC) from Wasabi Is a Promising Candidate for the Treatment of Cancer, Alzheimer’s Disease, and Obesity

**DOI:** 10.3390/nu16152509

**Published:** 2024-08-01

**Authors:** Joanna Bartkowiak-Wieczorek, Michał Malesza, Ida Malesza, Tomasz Hadada, Jakub Winkler-Galicki, Teresa Grzelak, Edyta Mądry

**Affiliations:** 1Physiology Department, Poznan University of Medical Sciences, 6, Święcickiego Street, 60-781 Poznan, Poland; mmalesza6@gmail.com (M.M.); thadada@ump.edu.pl (T.H.); jwinklergalicki@ump.edu.pl (J.W.-G.); tgrzelak@ump.edu.pl (T.G.); emadry@ump.edu.pl (E.M.); 2Department of Pediatric Gastroenterology and Metabolic Diseases, Poznan University of Medical Sciences, 61-701 Poznan, Poland; ida.malesza@gmail.com

**Keywords:** breast cancer, colorectal-stomach cancer, Parkinson’s disease, chronic fatigue syndrome, PPAR, AMPK, MAPK, mTOR, Nrf, ERK

## Abstract

Methylsulfinyl hexyl isothiocyanate (6-MSITC) isolated from *Eutrema japonicum* is a promising candidate for the treatment of breast cancer, colorectal and stomach cancer, metabolic syndrome, heart diseases, diabetes, and obesity due to its anti-inflammatory and antioxidant properties. Also, its neuroprotective properties, improving cognitive function and protecting dopaminergic neurons, make it an excellent candidate for treating neurodegenerative diseases like dementia, Alzheimer’s, and Parkinson’s disease. 6-MSITC acts on many signaling pathways, such as PPAR, AMPK, PI3K/AKT/mTOR, Nrf2/Keap1-ARE, ERK1/2-ELK1/CHOP/DR5, and MAPK. However, despite the very promising results of *in vitro* and *in vivo* animal studies and a few human studies, the molecule has not yet been thoroughly tested in the human population. Nonetheless, wasabi should be classified as a “superfood” for the primary and secondary prevention of human diseases. This article reviews the current state-of-the-art research on 6-MSITC and its potential clinical uses, discussing in detail the signaling pathways activated by the molecule and their interactions.

## 1. Introduction

Most human lifestyle-related diseases are associated with prolonged stress, excessive consumption of sugars and fats, lack of physical activity, or air pollution, with most originating from oxidative stress and prolonged low-grade inflammation [1,2]. Obesity, insulin resistance, cardiovascular disorders, metabolic syndrome, diabetes, cancers, allergies, and the degeneration of the central nervous system (such as Parkinson’s disease and Alzheimer’s disease) are highly prevalent, causing a huge healthcare system burden [3]. Therefore, the search for remedies limiting these diseases, especially natural products with few or no side effects and pleiotropic mechanisms of action, is important. They fit into Hippocrates’ still valid challenge, “Let food be thy medicine and medicine be thy food”.

Wasabi (*Eutrema japonicum*), particularly due to its active ingredient 6-methylsulfinyl hexyl isothiocyanate (6-MSITC), represents a promising example of a natural compound with a wide range of biological activities. It possesses strong anti-inflammatory, anti-atherosclerotic, and antioxidant properties, which can support cardiovascular function and are neuroprotective [4,5,6,7,8,9,10,11,12]. Also, it has a minimal toxicity risk compared with synthetic substances, making it an attractive candidate for the prevention and treatment of lifestyle-related diseases. This review systematically compiles current research on the molecular and biochemical mechanisms of the pleiotropic effects of wasabi and its active compound, 6-MSITC, in the prevention and therapy of human diseases.

### Search Strategy and Selection Criteria

A comprehensive search of the PubMed/Medline and Cochrane databases was conducted to retrieve relevant articles related to wasabi, wasabi extract, and 6-MSITC. The search utilized English terms and their combinations, including wasabi, wasabi extract, 6-MSITC, cancer, neurodegenerative diseases, Alzheimer’s disease, obesity, inflammation, oxidative stress, molecular pathways of the cell cycle, and isothiocyanate, and yielded 207 papers for “wasabi”, 27 papers for “6-MSITC”, 23 papers for “wasabi”and “6-MSITC”, and 23 papers for “6-MSITC” AND “wasabi” AND “isothiocyanate”. Specific combinations yielded the following results: (6-MSITC) AND (wasabi) AND (cancer) resulted in nine papers, (6-MSITC) AND (wasabi) AND (Parkinson) resulted in one paper, (6-MSITC) AND (wasabi) AND (Alzheimer) resulted in zero papers, (6-MSITC) AND (wasabi) AND (obesity) resulted in one paper, (6-MSITC) AND (wasabi) AND (neurodegenerative diseases) resulted in one paper, (6-MSITC) AND (wasabi) AND (inflammation) resulted in two papers, (6-MSITC) AND (wasabi) AND (oxidative stress) resulted in two papers, and (6-MSITC) AND (wasabi) AND (pathways of cell cycle) resulted in one paper. The titles and abstracts were screened to identify relevant articles for further review.

## 2. The Chemical and Biological Structure of 6-MSITC

6-MSITC is a derivative of glucosinolates present in the Brassicaceae family [5,6]. It is hydrolyzed by myrosinase to form hydroxamate-O-sulfonate, subsequently generating a variety of compounds, including nitriles, goitrin, epithionitriles, thiocyanates, and isothiocyanates [5,6,7]. 6-MSITC, along with allyl isothiocyanate (AITC), is the primary isothiocyanate constituent of wasabi, with additional members such as 6-methylthiohexyl isothiocyanate (6-MTITC) also present [8]. It is an analogue of sulforaphane allyl isothiocyanates, representing a significant allyl isothiocyanate, and is a potent analogue of sulforaphane [9].

The chemical structure of 6-MSITC is characterized by a linear alkyl chain with six carbon atoms, denoted as “hexyl”, and the presence of an isothiocyanate (ITC) group, which is represented as “–N=C=S” (Figure 1). Additionally, a methylsulfinyl group (CH_3_–S(=O)–) is attached to the hexyl chain. The ITC group, with a nitrogen atom at one end and a sulfur atom at the other, imparts its characteristic bioactivity. This group is crucial for the biological effects and reactivity of isothiocyanates, allowing them to readily conjugate with cellular molecules such as reduced glutathione (GSH). The methylsulfinyl group is an important structural element that consists of a methyl group (CH_3_) attached to a sulfinyl moiety (S(=O)), introducing a sulfur atom in an oxidized state. The presence of this methylsulfinyl group influences the cell membrane permeability of 6-MSITC [10]. 

## 3. Cellular Location and Cytological and Biological Significance of 6-MSITC

6-MSITC, as an isothiocyanate, exhibits a specific cellular localization in organisms. It penetrates cells through the cell membrane via diffusion and can traverse the cellular barrier due to its chemical structure, which may have significant implications for their interactions with cells and intracellular structures. Isothiocyanates, including 6-MSITC, are primarily metabolized through the mercapturic acid pathway [11], a process involved in eliminating chemical compounds from the body and influencing their biological activity. 6-MSITC can rapidly combine with intracellular GSH through the isothiocyanate group (–N=C=S) to affect oxidative processes and oxidative stress in cells.

Structural elements of 6-MSITC, such as the methylsulfinyl group (CH_3_–S(=O)–) and the length of the alkyl chain, are essential for cell membrane permeability, as they influence the substance’s ability to penetrate target cells [12]. 

## 4. LADME (Liberation, Absorption, Distribution, Metabolism, Excretion/Elimination) of 6-MSITC

The isothiocyanate 6-MSITC belongs to a class of biologically active compounds that share similar pharmacokinetic characteristics [12]. The limited availability of literature focusing on the pharmacokinetics of 6-MSITC necessitates an analogy-based exploration of its pharmacological activity.

In the broader context of isothiocyanates, extensive research has elucidated the intricacies of their metabolism, particularly the first and second phases of biotransformation. Cytochromes P450 (CYPs) have emerged as pivotal actors in the intricate biotransformation of carcinogenic xenobiotics [13]. Specifically, the CYP1 family, comprising subfamilies A and B, and the enzymes CYP1A1, CYP1A2, and CYP1B1, along with the CYP2E subfamily housing CYP2E1, significantly contribute to the bioactivation of chemical carcinogens [13]. Dietary exposure to isothiocyanates has been established as a modulator of the expression and activity of these enzymes [14]. 

Isothiocyanates, including 6-MSITC, have garnered attention as potent inducers of detoxification enzyme systems, such as quinone reductase, glutathione S-transferase, epoxide hydrolase, and UDP-glucuronosyl transferase [15]. 6-MSITC reduced the protein levels of phase I enzymes, thereby demonstrating inhibitory effects on the development of rat colonic cancer [16]. Furthermore, 6-MSITC decreased hepatic CYP1A2, -2B2, -2E1, and -3A2 by 7–44% compared with the vehicle control [17]. 

Prior studies have identified the involvement of specific hepatic CYPs, such as CYP1A1/2, CYP2B1/2, and CYP3A2, in the mutagenic activation of various carcinogens [18]. Additionally, CYP2B1/2 and -2E1 have been implicated in the metabolic activation of environmental N-nitrosamines into ultimate carcinogens [19]. The decreased levels of CYPs induced by 6-MSITC are anticipated to reduce and/or slow the transformation of metabolites of certain carcinogens to proximate or ultimate carcinogens. In contrast, 6-MSITC did not significantly alter UDP-glucuronosyl transferase (UDPGT) activity towards bilirubin, 4-nitrophenol, and testosterone in hepatic microsomes. Although glucuronidation is generally considered a detoxification reaction in the metabolism of various chemicals, including carcinogens, hepatic UDPGT activity is not a focal point of modification induced by 6-MSITC [20]. 

The pharmacological activity of 6-MSITC involves the modulation of phase I enzymes, particularly CYPs, which play a crucial role in the bioactivation of carcinogenic xenobiotics. This modulation is expected to contribute to the chemopreventive effects of 6-MSITC, potentially inhibiting the development of colonic cancer. Additionally, 6-MSITC induces detoxification enzymes, providing further evidence of its potential therapeutic benefits. These detailed insights deepen our understanding of the pharmacological actions of 6-MSITC and its integral role in xenobiotic metabolism [13,14,15,16,18,19,20].

## 5. Involvement of Wasabi Extracts/6-MSITC in Cellular Signaling Pathways

The nomenclature for signaling pathways responsible for cell metabolism and survival is highly varied. Many receptors and proteins participate in interconnected signaling pathways, and the analysis of a specific receptor’s regulation or the impact of its activation on other cellular processes is referred to as a signaling pathway. In this literature review, we focus on the most commonly described signaling pathways in scientific literature that may interact with and overlap with each other. It is possible that in other publications, elements of the signaling pathway described will be considered a separate signaling system or that the pathway presented is part of a larger, more comprehensive system. 6-MSITC is involved in modulating numerous signaling pathways by influencing receptors, kinases, and regulatory elements in DNA, thereby affecting their activity, as shown in Figure 2 and Table 1.

### 5.1. PPAR Signaling Pathway

PPAR (peroxisome proliferator-activated receptor) is a family of nuclear receptors of steroid origin that function as ligand-dependent transcription factors [46]. These receptors regulate the metabolism of fats, carbohydrates, and proteins, as well as cell division and immune processes, such as the intensification of inflammation [47]. Currently, three different types of these receptors have been identified: PPARα (also signified as NR1C1), PPARβ/δ (also signified as NR1C2), and PPARγ (also signified as NR1C3) [48]. 

PPARγ is a nuclear receptor crucial for adipocyte differentiation, lipid metabolism, and adipokine synthesis, playing a central role in metabolic disorders such as obesity and insulin resistance. It has been shown that wasabi extracts or the active ingredient of wasabi, 6-MSITC, inhibited adipose cell differentiation in both *in vivo* and *in vitro* studies. Ogawa et al. reported that water wasabi leaf extract decreased PPARγ, thereby downregulating the expression of adipogenic genes like *LPL*, *SCD1*, *ACC1*, *aP2*, *CD36*, *FAS*, and *PEPCK*. Consequently, wasabi leaf extract inhibited the hypertrophy of adipose tissue due to its 6-MSITC activity [49]. In a rat model of the metabolic syndrome, wasabi extract (4 g/kg/day) reduced PPARγ mRNA expression [50]. 

PPARα functions as a specific regulator of lipid metabolism, influencing gene expression in response to fatty acids. It promotes fatty acid oxidation and regulates lipoprotein metabolism, and its activity is enhanced by fatty acids. Therefore, it is engaged in fatty acid beta-oxidation and energy homeostasis [51]. In HepG2 cells, 6-MSITC decreased the level of PPARα and consequently inhibited lipid peroxidation, reducing overall oxidative stress via the AMPKα-PPARα/FOXO1-Sirtuin1 pathway [43].

### 5.2. AMP-Activated Protein Kinase (AMPK) Signaling Pathway

AMP-activated protein kinase (AMPK) is one of the most important regulators of cell energy metabolism. It is activated by decreasing ATP concentration and establishes cellular energy homeostasis by promoting glucose uptake and fatty acid oxidation and reducing energy demand. The AMPK signaling pathway is involved in communication between metabolically dependent tissues and hypothalamic hormones and cytokines/hormones secreted by adipose tissue (leptin and adiponectin) [52]. Oowatari et al. showed that rats treated with wasabi leaf extracts (4 g/kg/day) had increased levels of phosphorylated AMPK protein (pAMPKα and pAMPKβ), which enhances insulin sensitivity and has a positive impact on lipid metabolism. AMPK activation promotes hepatic fatty acid oxidation, prevents hepatic steatosis, and contributes to overall metabolic improvements. Biomarkers of liver injury (AST and ALT) and plasma triglycerides were lower in rats treated with wasabi leaf extract, indicating improved hepatic metabolism through adiponectin-AMPK signaling and the hepatoprotective activity of 6-MSITC [50]. 

Pan et al. analyzed the cytoprotective and antioxidant effects of 6-MSITC *in vitro*, demonstrating that 6-MSITC (10 μM) activated AMPK in a concentration-dependent manner in HepG2 cells. AMPK activation subsequently triggered a cascade of changes, including the increased expression of antioxidant enzymes such as FOXO1 and Sirtuin1 and the stimulation of the AMPKα/Nrf2 signaling pathway [43]. 

AMPK and NRF (Nuclear Respiratory Factor) are both involved in the regulation of cellular metabolism. Acting as an energy sensor, AMPK responds to decreased ATP levels, while NRF governs mitochondrial functions and biogenesis. The connection between AMPK and NRF lies in the potential impact of AMPK activation on mitochondrial processes, influencing gene expression regulated by NRF. This overlapping of molecular pathways establishes a cellular regulatory system that adapts metabolism and mitochondrial function to energy demands. Thus, 6-MSITC exhibits cytoprotective effects against metabolic lipid stress through AMPKα/Nrf2-mediated pathways [43]. 

### 5.3. The Nrf2/Keap1-ARE Pathway

Nrf2/Keap1-ARE is a complex pathway that serves as a defense mechanism against oxidative stress. Under normal conditions, Keap1 (Kelch-like ECH-associated protein 1) acts as a facilitator, overseeing the degradation of Nrf2 (nuclear factor erythroid 2-related factor 2). However, oxidative stress disrupts this process, leading to the accumulation of Nrf2 in the cell nucleus [31,32]. As a transcription factor, Nrf2 activates antioxidant genes by binding to their antioxidant response element (ARE) in promoter regions, such as glutathione S-transferase (*GST*), NAD(P)H quinone oxidoreductase 1 (*NQO1*), heme oxygenase 1 (*HO-1*), and γ-glutamylcysteine synthetase (*γ-GCS*) [53]. Under physiological conditions without oxidative stress, Nrf2 is regulated and inhibited by the protein Keap1 [54,55].

In the IMR-32 neuronal cell line, 6-MSITC increased Nrf2 protein levels but did not affect Keap1. Nrf2 activation led to its stabilization at the post-transcriptional level and enhanced expression of Nrf2-dependent proteins such as NQO1, TXNRD1, AKR1C1, and AKR1C3, demonstrating a neuroprotective effect *in vitro* [31,32]. 

In HepG2 cells, 6-MSITC, through its modulation of canonical pathways, notably the Nrf2-mediated oxidative stress response pathway, impacts the regulation of genes associated with glutamate metabolism, glutathione metabolism, and the NOTCH signaling pathway [37]. The NOTCH pathway is a highly conserved cell signaling system of great importance that can be found in most animals. It is crucial in development as it is involved in cell fate regulation, cell proliferation, and cell death. Moreover, NOTCH is implicated in malignant transformation [56]. In zebrafish larvae, 6-MSITC decreased arsenite toxicity via Nrf2-dependent antioxidant mechanisms [36]. Moreover, 6-MSITC exerts its influence on the ubiquitination and proteasomal turnover of Nrf2. The augmentation of Nrf2-ARE binding and transcription activity, coupled with the up-regulation of NQO1, highlights the intricacies of 6-MSITC’s action in cellular defense and cancer chemoprevention [26].

Furthermore, 6-MSITC treatment of HepG2 cells enhances aldehyde dehydrogenase (ALDH) activity by selectively inducing mitochondrial ALDH2 expression. The induction of ALDH2 depends on the presence of Nrf2, as evidenced by a reduction in ALDH2 expression following Nrf2 knockdown. 6-MSITC induced the nuclear translocation of Nrf2, concurrently upregulating the expression levels of HO-1 and SOD2, both of which are phase II drug-metabolizing enzymes regulated by Nrf2. Upon oral administration to C57BL/6J mice, 6-MSITC augments mitochondrial ALDH2 activity and expression in the liver [37], indicating that the hepatoprotective effect of 6-MSITC is partially mediated by Nrf2.

The complexity of these cellular processes reflects the intricate nature of Nrf2’s role in maintaining cellular homeostasis, thus underscoring the potential impact of 6-MSITC on cell functions [32].

### 5.4. ERK1/2-ELK1/CHOP/DR5 Pathway

The ERK1/2-ELK1/CHOP/DR5 pathway plays a pivotal role in promoting cell viability by inhibiting apoptosis [57]. Extracellular signal-regulated protein kinases 1 and 2 (ERK1/2) are MAPK kinases activated by another signaling pathway of kinases Ras→Raf→MEK1/2→ERK1/2. ERK1/2 activation is caused by upstream kinases such as RAF and MEK, leading to the phosphorylation of downstream effectors, including ELK1 and CHOP, which are transcription factors involved in cell survival and apoptosis. In the context of the DR5 receptor, a member of the TNF receptor superfamily, ERK1/2-mediated signaling can influence its expression and activation, further modulating the cellular response to apoptotic stimuli [58,59]. 

*In vitro* studies demonstrated that 6-MSITC induced apoptosis in human colorectal cancer cells by arresting the cell cycle in the G2/M phase via the ERK1/2-mediated ELK1/CHOP/DR5 pathway and in a p53-independent manner. In these studies, 6-MSITC caused stimulation of the ERK1/2 pathway, including ERK1/2 and ELK1 phosphorylation, and upregulation of CHOP and DR5 in the human colorectal cancer cell lines HCT116 *p53*^+/+^ [33,35]. 

### 5.5. The MAPK Pathways

The mitogen-activated protein kinase (MAPK) signaling pathways are responsible for the transmission of signals within cells and for inducing mitosis; therefore, they participate in oncogenesis, tumor progression, and the regulation of inflammatory responses. MAPK pathways engage among other protein kinases like ERK1/2, c-Jun N-terminal kinase (JNK1/2/3), p38 MAPK (α, β, γ, δ), and ERK5 [60,61]. The MAPK pathways are involved in the regulation of the expression of inflammatory genes such as cyclooxygenase-2 (*COX-2*) and nitric oxide synthase (iNOS) [62]. It has been shown that 6-MSITC inhibits LPS-induced phosphorylation of MAPKs and MAPKKs, thereby blocking COX-2 expression and selectively suppressing iNOS expression by targeting JNK phosphorylation [23,62]. 

In physiological conditions, the activity of AP-1 is very weak, but inflammatory factors increase its activity. Activated AP-1 via promotor elements stimulates the expression of inflammatory genes [63]. 6-MSITC also inhibits phosphorylation of c-Jun, which, together with FOS, forms activating protein 1 (AP-1). Therefore, by blocking c-Jun phosphorylation, 6-MSITC inhibits AP-1 activity, reducing the expression of inflammatory genes regulated by this protein complex [12]. 

As mentioned above, MAPK pathways also play an important role in controlling cell division. AP-1 and c-JUN are well-established oncogenic factors involved in the development of multiple cancers, but the significance of 6-MSITC in this field is yet to be determined [61]. 

### 5.6. PI3K/AKT/mTOR Pathway 

The phosphoinositide 3-kinase (PI3K)/serine-threonine kinase Akt signaling pathway is a key player in cancer development and progression. It is involved in regulating the cell cycle and programmed cell death. Akt also phosphorylates multiple protein targets such as phosphatase and tensin homolog (PTEN), Akt, TSC1, and mechanistic targets of rapamycin (mTOR) [64]. Moreover, the Akt pathway is pivotal in the regulation of transcriptional activity of nuclear factor-κB (NFκB), which is essential for PI3K/AKT/mTOR oncogenic activity [65]. 

In BALB/c female nude mice with MDA-MB-231 or -453 cells, 6-MSITC decreased the level of phosphorylated AKT in a dose-dependent manner, thus inhibiting the NF-κB pathway and promoting the apoptosis of breast cancer cells [30]. 

### 5.7. Inflammatory and Anti-Inflammatory Pathways

The anti-inflammatory activity of 6-MSITC was demonstrated by inhibiting COX-2 expression and PGE2 production in murine macrophages and human monocytic cells through signaling pathways and associated gene promoters (*NF-κB*, *C/EBP*, and *CRE*). Additionally, 6-MSITC inhibited NO synthesis by suppressing iNOS expression and modulating the expression of inflammatory cytokines, including IL-1β, IL-6, TNF, and interferon-inducible genes (*IFI1* and *IFI47*), as well as interleukin receptors induced by LPS. 6-MSITC restored the levels of CC chemokines and IL-3 and reduced the activity of JAK-STAT kinases, C/EBPδ expression, and JNK-mediated AP-1 activation. These effects highlight 6-MSITC’s potential to effectively suppress inflammatory pathways [12]. 

Moreover, water wasabi extract in murine cell line 3T3-L1 reduced the activity and aggregation of pro-inflammatory adipogenesis markers in a dose-dependent manner (glycerol-3-phosphate dehydrogenase and triglyceride) and decreased the expression of adipogenic genes like *LPL*, *SCD1*, *ACC1*, *aP2*, *CD36*, *FAS*, and *PEPCK* [49]. 

6-MSITC decreased pro-inflammatory (*TNFα*, *IL1β*, *IL6*, *PTGS*, *COX-2*) and interferon-inducible (*IFI1*, *IFI47*), and IL receptor (*IL10ra*, *IL23r*, *IL4ra*) genes and restored reduced anti-inflammatory genes (*Ccl11*, *Ccl25*, *IL3*, *IL1ra12*, *IL8ra*, *TNFRSF23*, *TNFRSF4*) [25]. Additionally, 6-MSITC suppressed IL-6 and CXCL10 production in TNF-α-treated human oral epithelial cells (TR146 cells) by inhibiting the activation of STAT3 and NF-κB pathways [24,41]. Anti-inflammatory activities of 6-MSITC are presented in Table 1. 

## 6. Other Activities of 6-MSITC

6-MSITC demonstrated analgesic activity by activating TRPA1 (transient receptor potential ankyrin 1) and TRPV1 (transient receptor potential vanilloid 1) channels, affecting pain perception and cellular responses in a manner dependent on the increase in Ca^2+^ concentration, suggesting similarity to other TRPA1 agonists [27]. TRPA1, a nonselective cation channel found in humans, is present in primary sensory neurons and is crucial for detecting pain stimuli triggered by cold, mechanical stimuli, intracellular alkalization, Ca^2+^ and Zn^2+^ ions, and pungent chemicals [66].

Additionally, 6-MSITC inhibited type I allergies and suppressed the release of histamine, leukotriene B4 (LTB4), and cysteinyl leukotrienes (CysLTs) by interfering with the intracellular Ca^2+^ elevation [67]. 

All the activities mentioned above, modulated by wasabi extract and 6-MSITC, are presented in Table 1. 

## 7. 6-MSITC and Cancer

Wasabi exhibits anti-cancer properties, attributed in part to the presence of 6-MSITC, which plays a crucial role in regulating various cellular processes, particularly those associated with cancer development and progression. Research on the anti-cancer effects of both Wasabi and its active component, 6-MSITC, yields promising results in the realm of pharmacological anti-tumor activity [12,30,32,35,43,50,68]. 

### 7.1. Cancer Cell Lines

*In vitro* analyses of the anticancer activity of 6-MSITC confirm the involvement of numerous pro-apoptotic signaling pathways. In HepG2 cells, 6-MSITC exerted a chemopreventive effect through its underlying antioxidant activity via the activation of Nrf2-mediated (see Section 5.3) subsequent induction of cytoprotective genes [32]. The antioxidant activity of 6-MSITC also prevents overloaded lipid stress in HepG2 cells by activating AMPKα (see Section 5.2) and Nrf2. It enhances the expression of Forkhead box protein O1 (FOXO1) and Sirtuin1, along with Nrf2 target proteins NAD(P)H, quinone oxidoreductase 1 (NQO1), and heme oxygenase (HO-1). 6-MSITC effectively reduces thiobarbituric acid-reactive substances and fat accumulation induced by CFA [26,43]. In RL34 cells, MSITC activated the Nrf2/ARE-dependent detoxification pathway (see Section 5.3) [21,68]. Furthermore, 6-MSITC induces apoptosis in human colorectal cancer cell lines (HCT116) by activating the ERK1/2-mediated ELK1/CHOP/DR5 pathway (see Section 5.4) [33,35], inhibits LPS-induced phosphorylation of MAPKs (ERK, p38 kinase, and JNK) (see Section 5.5), and regulates inflammatory responses comprehensively in several cell lines (see Section 5.7) [12]. 6-MSITC also mediated apoptotic effects through the PI3K/AKT pathway (see Section 5.6), causing a reduction in phosphorylated AKT and inhibiting NF-κB in MCF-7, MDA-MB-231, MDA-MB- 435S, and Hs578T human breast cancer cell lines [30]. 

### 7.2. Colorectal and Stomach Cancer Studies

In the broader context of public health, colorectal cancer and stomach cancer present significant challenges, ranking among the leading causes of cancer-related mortality. A comprehensive exploration of the complex relationship between dietary factors and the development of colorectal and stomach cancers is crucial for advancing therapeutic interventions in these distinct yet interconnected oncological domains [69]. Research on the impact of 6-MSITC as a chemopreventive and anticancer agent has been conducted not only on cell lines and animal models but also through clinical observations.

Yano et al. investigated the anti-cancer properties of 6-MSITC on human colorectal cancer cells (HCT116 p53^+/+^ and HCT116 p53^−/−^) to explore the anticancer activity and molecular mechanisms of 6-MSITC. Notably, 6-MSITC hindered cell proliferation in both cell types despite observing a slight increase in the phosphorylation and accumulation of the P53 protein in HCT116 p53^+/+^ cells 24 h after treatment. 6-MSITC induces apoptosis in human colorectal cancer cells through a p53-independent mitochondrial dysfunction pathway. Additionally, 6-MSITC-induced cell cycle arrest in the G2/M phase and apoptosis in both cell types stimulated ERK1/2 phosphorylation to activate the ERK1/2 pathway (see Section 5.4), including ELK1 phosphorylation, and upregulated C/EBP homologous protein (CHOP) and death receptor 5 (DR5) [33,35]. 

Wasabi reduced the incidence of forestomach epidermoid cysts and duodenal carcinosarcomas in a rat model of N-methyl-N′-nitro-N-nitrosoguanidine (MNNG)-induced gastric carcinogenesis, which encompassed adenocarcinomas, adenomatous polyps, adenomatous glandular hyperplasias, and duodenal adenocarcinomas. Additionally, wasabi exhibited a preventive effect on the development of stomach carcinogenesis. The underlying mechanism for wasabi’s anti-cancer efficacy is intricately linked to its bioactive compound, allyl isothiocyanate. Allyl isothiocyanate functions by inhibiting H+K+-ATPase and suppressing stomach acid secretion, thereby leading to a reduction in gastric pH and an alkalinization of the mucosal surface. This dual action of allyl isothiocyanate contributes significantly to its observed inhibitory impact on gastric carcinogenesis [70]. 

In a rat-based study, 6-MSITC demonstrated chemopreventive effects in short-term colon carcinogenesis by inhibiting epithelial cell proliferation and reducing the levels of drug-metabolizing cytochrome P-450 isozymes. Administration of 6-MSITC at 400 ppm resulted in a significant reduction in mucosal lesions, including total aberrant crypt foci (ACF), larger ACF, and β-catenin-accumulated crypts (BCAC), as well as a decrease in the size (crypt multiplicity) of BCAC. Immunohistochemically, 6-MSITC administration lowered the proliferating cell nuclear antigen labeling index in both ACF and BCAC [16]. 

### 7.3. Breast Cancer Studies

In a murine model, 30 female Balb-nu/nu mice inoculated with MDA-MB-231 or -453 breast cancer cells were subjected to oral administration of varying concentrations of 6-MSITC for 12 days following tumor growth. Orally administered 6-MSITC reduced tumor volume and mass in mice inoculated with MDA-MB-231 breast cancer cells, irrespective of estrogen receptor (ER) expression levels. Additionally, a concentration-dependent inhibitory effect of 6-MSITC on the antiapoptotic activity of NF-κB was observed in ER-negative MDA-MB-231 and MDA-MB-453 cells. The indirect inhibition of cyclin D1 by 6-MSITC, achieved through NF-κB inhibition, synergistically suppressed breast cancer growth by inhibiting arterialization or cell cycle progression. In MDA-MB-231 cells, 6-MSITC downregulated the expression of phosphorylated AKT and inhibited the PI3K/AKT pathway (see Section 5.6). These findings suggest that 6-MSITC may impede the development and progression of tumorigenesis *in vivo* by inhibiting proliferation or inducing apoptosis [30]. 

### 7.4. Metastasis Studies

The impact of orally administered 6-MSITC or a T-wasabi fraction from *Eutrema japonicum* on macroscopic pulmonary metastasis was investigated using a murine B16-BL6 melanoma model. The experiment involved two administration routes (subcutaneous or intravenous) and times (before or concomitant with tumor inoculation) of 6-MITC or T-wasabi, showing a significant reduction in metastasized foci per lung with either subcutaneous or intravenous injection of 6-MITC or the T-wasabi fraction. The maximum reduction achieved by the T-wasabi fraction was 82%. Moreover, a 2-week 6-MITC (200 μM) pre-treatment inhibited foci formation, while concomitant administration with tumor inoculation resulted in only 27% inhibition. No toxic effects were observed for 6-MITC or T-wasabi at the tested concentrations. The findings, in conjunction with previous results, suggest that a component of wasabi, 6-MITC, not only inhibits tumor cell growth but also suppresses tumor metastasis, making 6-MITC a potential dietary candidate for controlling tumor progression [71]. 

## 8. Neuroprotective Effects of 6-MSITC

In a 12-week double-blinded randomized controlled trial (RCT) involving seventy-two healthy older adults, the 6-MSITC group showed significant improvement in working and episodic memory performance compared with the placebo group. It has been suggested that the improvement of cognitive functions by 6-MSITC acts via the reduction of oxidative stress and inflammation in the hippocampus [44]. 

In a study by Oka et al., fifteen patients administered wasabi extract containing 6-MSITC (9.6 mg/day) for 12 weeks demonstrated improved cognitive function and performance in the Trail Making Test-A (TMT-A). Specifically, symptoms such as brain fog severity, difficulty in finding appropriate words, and photophobia significantly improved, and frontal lobe function was enhanced after 6-MSITC treatment. Additionally, psychological parameters, assessed through the vigor score, demonstrated substantial improvement, along with enhancements in health-related quality of life, general health perception, and vitality. Patients perceived the treatment as effective for cognitive issues and brain fog [40]. 

Morroni et al. subjected mice to 6-MSITC treatment (5 mg/kg twice a week) for four weeks following unilateral intrastriatal injection of 6-hydroxydopamine (6-OHDA), revealing the pronounced neuroprotective effects of 6-MSITC, particularly in preserving functional nigral dopaminergic neurons. This preservation was pivotal in mitigating motor dysfunction induced by 6-OHDA, reducing apoptotic cell death, and activating glutathione-dependent antioxidant systems. The administration of 6-MSITC significantly improved rotarod performance in the presence of 6-OHDA-induced motor deficits, showcasing its positive impact on motor behavior and coordination. Additionally, the treatment demonstrated efficacy in preserving dopaminergic neurons in the substantia nigra and striatum, underscoring its protective effects. Furthermore, 6-MSITC inhibited cell death and apoptosis, contributing to overall neuroprotection. Restoring redox status and GSH levels, critical components in cellular defense against oxidative damage, further emphasized the comprehensive neuroprotective actions of 6-MSITC. These findings highlight the multifaceted neuroprotective effects of 6-MSITC, encompassing improvements in motor performance, preservation of dopaminergic neurons, inhibition of cell death and apoptosis, and restoration of redox balance and GSH levels, underscoring the potential therapeutic significance of 6-MSITC in the context of Parkinson’s disease [29]. 

The influence of a wasabi root extract product enriched with 6-(methylsulfinyl) hexyl isothiocyanate, referred to as “Wasabi Sulfinyl (WS)”, on neurocognitive functions in cognitively intact elderly individuals was investigated through a randomized, double-blind, placebo-controlled trial. Fifty cognitively intact adults experiencing memory complaints (aged 45 to 69 years) were administered either 100 mg of WS or a placebo daily for 8 weeks. Favorable effects of WS were observed across nearly all parameters of the group version of the Stroop Color Word Test (G-SCWT), with no adverse effects related to WS consumption observed. The outcomes suggest that WS may positively affect attention- and processing-related cognitive performance in cognitively intact middle-aged and older adults. Post-hoc subgroup analyses further indicated that individuals not engaging in regular exercise were more likely to respond positively to WS [72]. 

The neuroprotective effects of 6-MSITC through the activation of the transcription factor Nrf2 (NF-E2-related factor 2) were investigated using the AppNL-G-F/NL-G-F knock-in (AppNLGF) mouse model of Alzheimer’s disease (AD). In the context of pharmacological Nrf2 activation, the administration of 6-MSITC improved cognitive impairments in AppNLGF mice in a passive-avoidance task. The study focused on administering 6-MSITC to mice orally, evaluating its potential to prevent cognitive decline [38]. 

Behavioral and pathological analyses revealed a significant improvement in escape latency in the passive-avoidance task for 6-MSITC-treated AppNLGF mice compared with those treated with a vehicle. In contrast, no significant change was observed in WT mice. Immunofluorescent staining for Aβ and Iba1 demonstrated that prolonged 6-MSITC treatment reduced the number of Iba1-positive microglia associated with amyloid plaques in the mouse cerebral cortex. The study further explored the ability of 6-MSITC to activate Nrf2 signaling (Section 5.3) in the brain. Long-term treatment tended to increase the expression of the *Nqo1* gene. Additionally, under acute induction conditions, high-dose 6-MSITC administration to WT mice resulted in elevated *Nqo1* mRNA expression in the cerebral cortex. These findings support the hypothesis that sustained treatment with mild Nrf2 inducers, such as 6-MSITC, may prevent the onset of cognitive impairment in a murine model of Alzheimer’s disease [38]. 

The influence of wasabi and 6-MSITC on cognitive functions is presented in Table 1. 

### 8.1. 6-MSITC in Alzheimer’s Disease

Alzheimer’s disease (AD) is a neurodegenerative disorder (NDD) characterized by irreversible changes in the brain leading to dementia. Neuropathological hallmarks of AD are deposits of amyloid β-peptide (Aβ), neurofibrillary tangles consisting of tau protein, and gradual loss of synapses. Hyperphosphorylation of tau protein negatively affects the clearance of cellular proteins, thus increasing Aβ generation via impairment of the lysosomal/autophagic pathway [73,74]. The level of tau hyperphosphorylation correlates with cognitive deficits in AD [75,76]. Genetic, cellular, and metabolic changes are considered to play a critical role in AD development, and oxidative stress (OS), endoplasmic reticulum stress (ERS), and neuroinflammation are thought to be the main drivers of the early stages of AD [76].

Contemporary AD therapies focusing on halting the progression of the disease are not particularly effective, so attention has turned to brain neuroprotection. 6-MSITC has been identified as a potential agent with neuroprotective potential [10,29].

Morroni et al. observed that intraperitoneal administration of 6-MSITC alleviated Aβ1-42O-induced memory impairment, oxidative stress, neuroinflammation, and hippocampal neuronal degeneration in mice after intracerebroventricular injections of Aβ1-42 peptide. Moreover, histological examination showed that 6-MSITC prevented neuronal death in the hippocampus [10]. Administration of 6-MSITC in Aβ mice led to several changes, including inhibition of excessive ROS production, potentially mediated by restoring glutathione levels in the hippocampus, which was depleted in mice treated with Aβ1-42O. Nrf2 (see Section 5.3), which regulates ARE-dependent transcription, a vital pathway in maintaining redox homeostasis, is affected in AD [10,34,77]. 6-MSITC treatment restored Nrf2 DNA binding. Uruno et al. showed that 6-MSITC induces Nrf2 and ameliorates AD-like pathology in mice [38]. 

The MAPK signaling pathway (see Section 5.5) is linked to neurodegeneration [78,79,80], hyperphosphorylation of tau [81], and neuronal apoptosis [10,80,81,82,83]. ERK1/2 (see Section 5.4), a member of the MAPK family, is associated with hippocampal caspase activation. 6-MSITC suppressed the phosphorylation of ERK1/2 in Aβ0-injected mice, thus reducing neuronal apoptosis [10]. The activation of oxidative stress-induced cell death biomarkers, caspases-9 and -3, was inhibited by 6-MSITC [10].

6-MSITC reversed glycogen synthase kinase-3β (GSK3β) phosphorylation caused by Aβ1-42O exposure [10,34,84]. GSK3β, a protein involved in the pathogenesis of AD, induces phosphorylation of the amyloid-β protein precursor and hyperphosphorylation of tau proteins [85,86,87]. Inhibition of GSK3β ameliorates the AD course and is regarded as a therapeutic target for novel drugs [85]. However, this requires further investigation [10,83].

Markers of activated astrocytes and microglia, reactive ionized calcium-binding adaptor molecule 1 (Iba-1) and glial fibrillary acidic protein (GFAP), were lowered by 6-MSTIC [10].

The inducible isoform of the iNOS, a hallmark of the classically activated pro-inflammatory phenotype in NDD, was lowered by 6-MSTIC in the Aβ-injected mice [10,88].

COX-2 is involved in the pathogenesis of AD and plays a vital role in inflammation. It is a common target for anti-inflammatory drugs, such as non-steroidal anti-inflammatory drugs [89]. The *COX-2* gene is mainly considered inducible, although Jung et al. showed that it is constitutively expressed in the neocortex and hippocampus in a murine model [90]. Induction of COX-2 via Aβ leads to neuroinflammation. Uto et al. showed that 6-MSITC suppressed LPS-induced COX-2 expression by blocking the MAPK-signaling pathway (see Section 5.5) in murine macrophages [12,22]. Presumably, 6-MSITC can inhibit COX-2 expression in the AD model, which requires further studies.

Whether these effects apply to humans remains unsolved, yet it is a potential premise for developing neuroprotective agents. 

### 8.2. 6-MSITC in Parkinson’s Disease

Parkinson’s disease (PD) is a progressive neurodegenerative disorder with cognitive impairment, mood and sleep disturbances, as well as autonomic dysfunction [91]. Loss of dopaminergic neurons due to abnormal protein aggregates of α-synuclein within neurons, known as Lewy bodies, causes a discrepancy in the activity between neuronal pathways in the basal ganglia, resulting in movement disorders [91,92]. Neurodegeneration in PD results from mitochondrial dysfunction, OS caused by ROS generated by dopamine metabolism, misfolded protein aggregates, neuroinflammation, and unrestrained apoptosis [93,94,95,96]. 

To date, only one study has explored the effect of 6-MSITC on PD. Morroni et al. investigated the impact of 6-MSTIC on a rodent model of PD by injecting mice with 6-hydroxydopamine into the stratum, inducing neurodegeneration similar to that in human PD [29]. 6-MSITC exerted neuroprotective properties, alleviating changes caused by 6-hydroxydopamine (6-OHDA) and decreasing neuronal apoptosis and oxidative stress. 6-MISTC counteracted OS by maintaining the intracellular level of GSH, probably owing to the increased activity of glutathione reductase. The 6-OHDA-affected mice treated with 6-MSITC showed significantly improved motor function compared with the control group [29].

As aforementioned, 6-MSITC neuroprotective effects could be ascribed to its possible regulation of the Nrf2/ARE detoxification pathway (see Section 5.3) [97]. 

### 8.3. 6-MSITC in Chronic Fatigue Syndrome

Myalgic encephalomyelitis (ME) or chronic fatigue syndrome (CFS) is a chronic neurological condition characterized by extreme tiredness, post-exertional malaise (PEM), orthostatic intolerance, and a low quality of sleep that further affects cognition and mood [98]. 

There is no specific treatment for ME/CFS. The current therapeutic approach focuses on introducing coping behaviors, such as energy management, to plan energy expenditure, decrease sensory stimulation in patients’ environments, and manage sleep issues. If these are insufficient, a pharmacologic approach is introduced.

Oka et al., in an open-label study, investigated the effects of 6-MSITC on a group of patients with ME/CFS, measuring the effectiveness of the wasabi compound on sleep, fatigue, and cognition, among other parameters, for 12 weeks. The study failed to report any positive changes regarding fatigue, but there was a significant improvement in cognition and pain control. As in previous studies, the researchers attribute the result to the inhibition of oxidative stress and neuroinflammation by 6-MSITC [40].

However, Tanabe et al. showed in a clinical study that the consumption of 6-MSITC did not reduce post-exercise muscle damage through a mechanism dependent on calpain-1 (the marker of continual strength loss after exercises [39]. 

Altogether, 6-MSITC is potentially a potent antioxidant compound and a promisingly safe supplement that could improve cognition and subjective feelings of fatigue in various clinical populations. However, there is a need to confirm the results of initial studies in randomized, controlled trials.

### 8.4. 6-MSITC and Cognitive Functions

Only a few studies have evaluated the effects of 6-MSITC on cognitive functions [40,44]. Nouchi et al. found that 6-MSITC has beneficial properties for healthy older adults’ episodic memory and working memory capacity [44]. In contrast, Oka et al. reported positive effects of 6-MISTC on brain fog in patients with myalgic encephalomyelitis/chronic fatigue syndrome (ME/CFS) [40]. Yet, no effects have been observed on inhibition or processing speed. Interestingly, another bioactive compound of wasabi, sulforaphane (or 4-MSITC), has been reported to improve processing and working memory performance in healthy older adults [99]. Nouchi et al. hypothesized that the precognitive effects of 6-MSITC could be attributed to its anti-inflammatory and antioxidant properties (Section 5.7) [44,99]. 

In another single-arm, open-label study, researchers reported that the antioxidant and mitochondrial activating effects of 6-MSITC might exhibit anti-fatigue effects [61]. Perceived fatigue, ability to recover, stress level, and sleep quality significantly improved after four weeks of 6-MSITC intake in healthy volunteers [45]. 

## 9. 6-MSITC and the Endothelium

Cytotoxicity and leukocyte adhesion pathways can be activated in physiological processes and pathological inflammation situations. The most common problem is leukocyte cell-to-cell adhesion to other cell types, such as vascular endothelial cells (ECs) [100]. In the presence of inflammatory stimuli, activated ECs induce tissue factor (TF) expression on the cell surface to initiate blood coagulation via the extrinsic clotting pathway [101]. Moreover, the activated ECs can also release von Willebrand factor (VWF), which binds to coagulation factor VIII [102]. Additionally, activated ECs increase the expression of intracellular adhesion molecule 1 (ICAM-1), vascular cell adhesion molecule 1 (VCAM-1), and monocyte chemoattractant protein-1 (MCP-1), leading to leukocyte adhesion and transmigration [103]. EC dysfunction causes the pathological activation of blood coagulation and inflammation [28]. 

It has been shown that 6-MSITC at concentrations ranging from 0 to 1 µg/mL may exert anticoagulant and anti-inflammatory properties (see Section 5.7). 6-MSITC reduced TF expression in activated HUVECs (human umbilical vein endothelial cells) and modulated the generation of activated protein C, which is essential for the negative regulation of blood coagulation in normal ECs. The anticoagulant properties of 6-MSITC were established by examining its effects on the expression of pro-coagulant genes in activated ECs. 6-MSITC inhibited thrombin-induced platelet aggregation through binding to the cysteine residue of platelet proteins [104]. No changes were observed in VWF, PT, and APTT levels, suggesting that 6-MSITC does not affect blood coagulation factors [28]. 

6-MSITC reduced TNF-α-induced IL-6 and MCP-1 expression, alleviated TNF-α-induced adhesion of human monoblast U937 cells to HUVECs, and reduced VCAM-1 and E-selectin mRNA expression in activated ECs [28]. 

## 10. 6-MSITC in Diet-Induced Metabolic Syndrome

Obesity and metabolic syndrome are associated with a risk of developing cardiovascular disease, type 2 diabetes, non-alcoholic fatty liver disease, and certain cancers [105]. 

The impact of wasabi on carbohydrate-lipid metabolism parameters observed *in vivo* was at the physiological level (reduction of adipose tissue hypertrophy, appetite suppression) [42,50], histological level (reduction of fat deposition in 3T3-L1 preadipocytes in the liver) [42], biochemical level (lowering glucose levels, inducing insulin secretion, reducing total cholesterol, triglycerides, and creatinine in the blood) [4,42] and molecular level. The appetite-suppressing and irritant effects of wasabi were mediated through TRPA1 and TRPV in rats (see Section 6) [66]. The inhibition of adipose tissue hypertrophy was achieved through the increase in adiponectin levels, suppression of PPARγ expression (see Section 5.1), and stimulation of AMPK (see Section 5.2) activity by increasing adiponectin levels [50]. Furthermore, there was an increase in the expression of glucose transporter-2, PPARγ, insulin receptor substrate-1, and Nrf2 (see Section 5.3) in the liver and kidneys, along with a decrease in NF-κB levels [4]. 

Moreover, wasabi, especially 6-MSITC, showed anti-inflammatory activity (see Section 5.7) through different mechanisms, including inhibition of the NF-κB pathway [25].

## 11. Discussion

Wasabi and 6-MSITC have pleiotropic mechanisms of action and exert anti-inflammatory, antioxidant, anti-atherosclerotic, and anti-carcinogenic effects. Therefore, Wasabi has broad health-beneficial properties and the potential to be used to treat important global lifestyle-related health issues, including metabolic and cardiovascular diseases, neurodegenerative diseases, and malignancies. Advancing our knowledge about these plant-derived substances demands rigorous clinical research to provide reliable assessments of their efficacy and safety in humans. Progress in this field is crucial for validating existing findings and exploring novel applications of Wasabi and 6-MSITC in the prevention and treatment of entities discussed in the article.

Despite the promising results of *in vitro* and *in vivo* animal studies and a few human studies, Wasabi and 6-MSITC are yet to be evaluated in clinical trials. The necessity for extensive clinical trials arises due to existing gaps in scientific knowledge, and while preclinical studies *in vitro* and *in vivo* provide insights into 6-MSITC mechanisms of action, the intricacies of human physiology and responses necessitate thorough clinical analysis.

The pleiotropic effects of Wasabi and 6-MSITC stem from their complex chemical composition, implicating potential interactions between components and requiring profound understanding. Clinical studies should encompass molecular mechanisms at the cellular level and, more importantly, the optimization of dosages, bioavailability, and pharmacokinetics.

Human studies with Wasabi and 6-MSITC, although limited, have shown promising results, but the variation in doses emphasizes the need for further research to determine optimal therapeutic ranges in clinical settings. For instance, administration of 9.6 mg/day 6-MSITC improved cognitive dysfunction symptoms in patients with chronic fatigue syndrome [40]. In another study, 6-MSITC at 20 mg/kg and 40 mg/kg reduced ALDH2 enzyme activity and protein expression, alleviating acetaldehyde-induced cytotoxicity in liver cancer cells [37]. In a clinical study involving eleven healthy volunteers, the consumption of 500 mg/day of Wasabi sulfinyl was well tolerated, with three adverse events in two out of eleven participants (18.2%) classified as unrelated to Wasabi intake [72].

The literature concerning the consumption of wasabi or its active compound, 6-MSITC, primarily highlights its health benefits. Isolated instances of adverse reactions have been detailed in the work of Oka et al., with a documented case report specifically addressing cardiotoxicity following wasabi consumption.

A notable incident involved the consumption of a substantial amount of wasabi, which led to Takotsubo cardiomyopathy in a 60-year-old woman. It was hypothesized that cardiomyopathy was a stress-induced reaction, resulting in heightened catecholamine levels and increased nervous system activity [106]. There were no signs of anaphylactic symptoms, and the patient had no history of heart disease or other significant illnesses, with no mention of concurrent medications she might have been taking [107]. 

Finkel-Oron [107] discussed cases of anaphylaxis following Wasabi consumption but no reports or descriptions of such incidents have been found in scientific databases like PubMed and Medline. 

Moreover, 6-MSITC is currently available on the Japanese market as a supplement to enhance cognitive function. No severe side effects associated with its use have been reported, which indicates that 6-MSITC can be considered safe to use overall [72]. The diverse administration methods used in these studies, including oral administration and dietary supplementation, highlight the versatility and ease of use of 6-MSITC as a therapeutic substance [16,27]. 

The synthetic drugs used to manage metabolic and cardiovascular diseases, neurodegeneration, and malignancies have multiple diverse side effects in contrast to Wasabi and its compound 6-MSITC, which are characterized by the absence of reported side effects. Due to their multifaceted health-promoting potential, Wasabi and 6-MSITC show promise for adjuvant therapeutic applications while maintaining a high safety profile. Nevertheless, their clinical efficacy compared with standard therapy still needs to be investigated.

Analysis of existing studies on 6-MSITC reveals several limitations and challenges that may affect this molecule’s comprehensive understanding and clinical utilization. One of the main challenges is the limited availability of detailed pharmacokinetic data concerning 6-MSITC. Thus, this molecule’s absorption, distribution, metabolism, and excretion processes in humans are not fully understood. Consequently, many conclusions are based on analogies to other isothiocyanates, which can lead to incomplete or inaccurate interpretations.

6-MSITC affects various molecular pathways, including the Nrf2/Keap1-ARE pathway and pathways related to glutathione metabolism and NOTCH signaling. The complexity of these mechanisms poses a challenge in accurately determining the specific biological effects of 6-MSITC and interactions between different signaling pathways. Many 6-MSITC studies have been conducted under *in vitro* conditions, which may only partially reflect its actions in living organisms. Although *in vitro* results provide valuable insights into potential mechanisms of action, further *in vivo* studies are necessary to confirm these observations and understand how 6-MSITC behaves in more complex biological systems.

While studies suggest potential health benefits and overall safety of 6-MSITC, there is limited data on its long-term toxicity and side effects. Without accurate toxicological data, assessing the safety of long-term use of 6-MSITC is challenging and remains a question of the future. Moreover, variations in response to 6-MSITC may arise from individual genetic and biological variability. For instance, differences in the expression of enzymes metabolizing 6-MSITC can affect its efficacy and safety in different populations, so studies addressing these variabilities are needed to develop personalized approaches to therapy.

Developing precise analytical methods to measure the concentration of 6-MSITC and its metabolites in various biological matrices is crucial. Standardization of these methods is necessary to ensure consistency and comparability of results across different research teams.

Therefore, understanding the potential of 6-MSITC in medicine requires further research that addresses these challenges. Future efforts should focus on expanding pharmacokinetic and toxicological studies, conducting more *in vivo* research to validate *in vitro* findings, developing personalized therapeutic strategies considering genetic variability, and ultimately standardizing analytical methods and research protocols.

## 12. Summary

Figure 3 summarizes the activity of the 6-MSITC from *Eutrema japonicum* in medically relevant biological processes, demonstrating its various health-promoting effects. The preclinical *in vitro* and *in vivo* studies on animals and the few clinical trials in humans have confirmed its anti-inflammatory, anti-allergic, and antioxidant activities. The anti-inflammatory effect is related to stimulating leukocyte adhesion, anti-inflammatory cytokines, and inhibiting pro-inflammatory cytokines such as TNF-α and nitric oxide. 6-MSITC also causes apoptosis in cancer cells, exhibiting anticancer properties confirmed for breast, stomach, and colorectal cancers and concerning cancer metastasis.

The molecule also demonstrates metabolic benefits by enhancing lipid metabolism and antithrombotic and antiplatelet activities, collectively contributing to its cardioprotective effects.

This compound also has antiallergic and pro-cognitive effects, protecting nerve cells from oxidative stress and reducing neuroinflammation, which is considered one of the fundamental processes responsible for neuron degeneration.

6-MSITC participates in pain perception by maintaining homeostasis in pathways responsible for conducting thermal and pain stimuli. Furthermore, it reduces oxidative and mitochondrial stress arising from oxidative processes in cells, thereby supporting the body’s overall homeostasis.

## 13. Concussions

Studies conducted so far indicate that compounds of wasabi (lac. *Eutrema japonicum*) have a variety of health-promoting properties. Its active ingredient, 6-methylsulfinyl hexyl isothiocyanate (6-MSITC), represents anti-inflammatory, anti-oxidant, anti-cancer, anti-atherogenic, and neuroprotective activities. Such a wide range of proven activities points out that 6-MSITC is a promising candidate for combating the most critical lifestyle diseases such as metabolic and cardiovascular, Parkinson’s and Alzheimer’s diseases, and malignancies. However, to date, the amount of information on the effects of 6-MSITC on the human body is limited, indicating the need for further well-designed studies in humans.

Future efforts should focus on expanding pharmacokinetic and toxicological studies, conducting more *in vivo* research to validate *in vitro* findings, and developing precise analytical methods to measure the concentration of 6-MSITC and its metabolites in various biological matrices.

## Figures and Tables

**Figure 1 nutrients-16-02509-f001:**
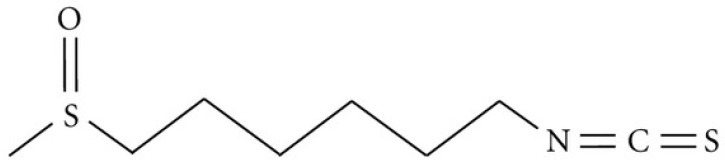
Structure of 6-MSITC.

**Figure 2 nutrients-16-02509-f002:**
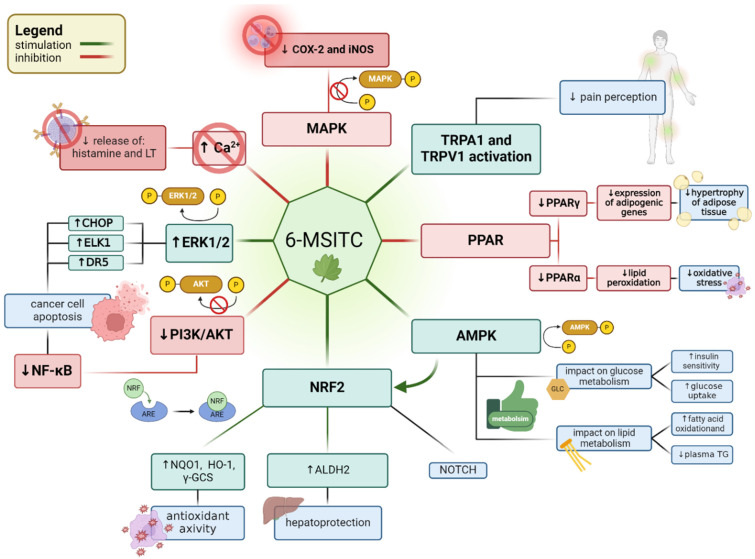
The influence of 6-MSITC on various signalling pathways (↑ increase; ↓ decrease).

**Figure 3 nutrients-16-02509-f003:**
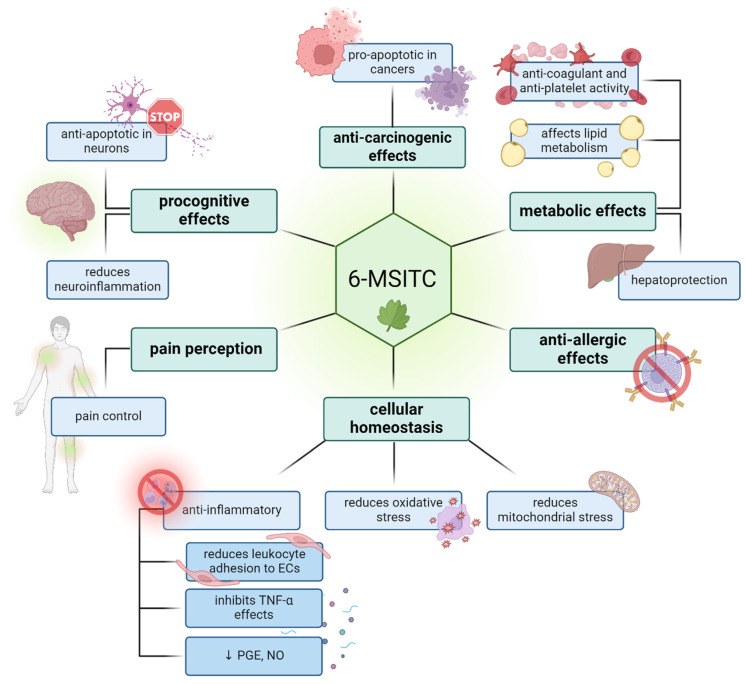
A summary of 6-MSITC biological activities (↓ decrease).

**Table 1 nutrients-16-02509-t001:** The chronological summary of research results on 6-MSITC molecule—*in vitro*, *in vivo*, and human studies.

	References/ Publication Year	Type and Duration of Experiment	Wasabi/6-MSITC Dose/Medium/Diet	Findings
1.	[21]/2002	*In vitro*/*in vivo*/5 days rat liver epithelial RL 34 cells	Eagle’a medium 15 μM per 5 days	6-HITC activated the antioxidant response element (ARE), 6-HITC induced the nuclear localization of the transcription factor Nrf2, which binds to the ARE, and the induction of phase II enzyme genes by 6-HITC was completely blocked in Nrf2 knockout mice. 6-HITC may be a potential activator of the Nrf2/ARE-dependent detoxification pathway.
		*In vivo*/7 days Male Wistar Rats	Control diet: 20% casein, 3.5% mineral (93G-MX), 5.0% vitamin (93-VX), 0.2% choline chloride, 5.0% corn oil, 4.0% cellulose powder, 22.1% sucrose, and 44.2% starch	
		*In vivo*/12 days Female ICR mice	Control diet: 20% casein, 3.5% mineral (93G-MX), 5.0% vitamin (93-VX), 0.2% choline chloride, 5.0% corn oil, 4.0% cellulose powder, 22.1% sucrose, and 44.2% starch 15 μM per 5 days	
2.	[22]/2005	*In vitro* Murine macrophage-like RAW264 cells	Dulbecco’s Modified Eagle’s Medium 16 μM 6-MITC. -6-MITC inhibits LPS-induced COX-2	6-MITC inhibited LPS-induced COX-2 expression at the signaling level and at the transcription factor/promoter levels.
3.	[22]/2005	*In vitro*/39 h Murine macrophage-like RAW264 cells	37 °C in a 5% CO_2_ atmosphere in Dulbecco’s modified Eagle’s medium containing 10% FBS.	6-MITC inhibited LPS-induced iNOS expression at the cellular signaling level. LPS induced iNOS expression by activating the Jak2-mediated JNK signaling cascade with the attendant AP-1 activation. 6-MITC blocked LPS-induced iNOS expression through the blockage of the Jak2 signaling cascade, leading to JNK-mediated AP-1 activation.
4.	[23]/2007	*In vitro*/30 h Murine macrophage-like RAW264 cells	Dulbecco’s Modified Eagle’s Medium 15 μM 6-MITC	6-MITC suppressed COX-2 expression induced by LPS or IFN-y, but not TPA, in murine macrophages. Molecular analysis revealed that LPS, INF-y, and TPA might induce COX-2 expression through different signaling pathways.
5.	[24]/2008	*In vitro*/17 h Murine macrophage-like RAW264 cells	8 μM 6-MSITC, 40 ng/mL LPS Dulbecco’s modified Eagle’s medium	6-MSITC may target immune and inflammation-related genes, including chemokines, interleukins, and interferons, to exert its anti-inflammatory function.
6.	[25]/2010	*In vitro*/21 h Murine macrophage-like RAW264	Dimethyl sulfoxide final concentration, 0.2% 8 μM 6-MSITC	The genes induced by 6-MSITC are CC chemokines (*CCL11* (C-C Motif Chemokine Ligand 11), *CCL25* (C-C Motif Chemokine Ligand 25), interleukins *IL3* (Interleukin 3) and receptors: *IL1ra12* (Interleukin 1 Receptor Antagonist 12), *IL8ra* (Interleukin 8 Receptor Alpha), *TNFRSF23* (Tumor Necrosis Factor Receptor Superfamily Member 23), *TNFRSF4* (Tumor Necrosis Factor Receptor Superfamily Member 4)
7.	[16]/2010	*In vivo*/12 weeks male F344 rats	Experiment 1 (sixty-six rats divided into seven groups): Groups 1–5: four weekly subcutaneous injections of DMH (40 mg/kg body weight) Groups 2–3: diet containing 200 and 400 ppm of 6-MSITC, respectively, for 5 weeks. Basic diet until the end of the study. Groups 4–5: mixed diet with 200 and 400 ppm 6-MSITC, respectively, from the first week after completion of treatment until the end of the study. Group 6: diet containing 400 ppm 6-MSITC throughout the study. Group 7: control	The dietary administration of 6-MSITC can significantly inhibit the induction of colonic AcF and BcAc by DMH by reducing cell proliferative activity and the protein levels of phase I enzymes.
			Experiment 2 (nine rats divided into three groups): Group 1: corn oil by gavage—control Groups 2–3: 6-MSITC at a dose of 20 and 40 mg/kg, respectively, in corn oil by gavage	
8.	[26]/2011	*In vitro*/12 h Human hepatoblastoma HepG2 cells	0–20 μM Dulbecco’s modified Eagle’s medium	6-MSITC regulated Nrf2-mediated ARE activation by targeting Nrf2 and Keap1. 6-MSITC reduced the level of Keap1 by modifying Keap1 and enhanced the level of Nrf2 by inhibiting Nrf2 ubiquitination and turnover. Finally, it resulted in a high ratio of Nrf2/Keap1. The surplus Nrf2, compared with Keap1, might bypass Keap1-Cul3 and accumulate in the nucleus to mediate ARE-driven activation.
9.	[27]/2012	*In vitro*/*in vivo*/10 min Male mice C57BL/6 (4–5 weeks; SLC), and TRPA1-deficient mice TRPA1-KO mice, Human embryonic kidney-derived 293 (HEK293) cells	20 μL 6-MSITC 10 or 30 mM Standard diet-free access Dulbecco’s Modified Eagle’s Medium	The results indicate the following points: (1) 6-MSITC and 6-MTITC activate both mTRPA1 and hTRPA1; (2) 6-MSITC activates mTRPV1; (3) The pharmacological functions of these isothiocyanates may result from TRPA1 activation.
10.	[28]/2014	*In vitro*/24 h Primary human umbilical vein endothelial cells (HUVECs)	0–30 ng/mL Endothelial Cell Growth Medium-2 BulletKit	The antiplatelet and anti-inflammatory effects of 6-MSITC on human umbilical vein endothelial cells (HUVECs) have been demonstrated. 6-MSITC slightly decreased tissue factor expression but did not affect von Willebrand release in activated HUVECs. 6-MSITC modified the generation of activated protein C, which is important for the negative regulation of blood coagulation in normal endothelial cells. 6-MSITC reduced the expression of interleukin-6 and monocytic chemotactic factor protein-1 induced by tumor necrosis factor alpha (TNFa). 6-MSITC significantly attenuated TNFa-induced adhesion of U937 monoblast cells to HUVECs and reduced mRNA expression of cellular vascular adhesion factor-1 and E-selectin in activated endothelial cells. 6-MSITC modifies EC function, inhibits cell adhesion, and 6-MSITC exerts anti-inflammatory effects, suggesting that it may have therapeutic potential as a treatment for vasculitis.
11.	[29]/2014	*In vivo*/4 weeks Male mice C57B1/6 (9 weeks old, 25–30g body weight at the beginning of the experiment) Animals were randomly divided into four groups (n 1/4 10–12 per group)	5 mg/kg 6-MSITC Twice a week Standard diet Free access	Administration of 6-MSITC for a month is able to exert neuroprotective effects in the 6-OHDA model of Parkinson’s disease. Treatment with 6-MSITC resulted in a significant reduction in oxidative stress and apoptotic cell death, leading to the improvement of behavioral disorders, especially motor deficits.
12.	[30]/2014	*In vitro*/*in vivo*/12 days *in vivo*: 30 female BALB/c female nude mice with MDA-MB-231 or -453 cells *in vitro*: breast cancer cell lines (MCF-7, MDA-MB-231, MDA-MB-435S, Hs578T, MDA-MB-453, BT-474, and DU4475	irradiated CE-7 basal diet—7 days 6.25 mg/kg, 25 mg/kg, and 100 mg/kg of 6-MSITC in sterile deionized water with the use of a stomach sonde needle for 5 days/wk.	The inhibitory effect of 6-MSITC on human breast cancer in a mouse model and the induction of apoptosis in human breast cancer by possible involvement of the NF-κB pathways have been revealed.
13.	[31]/2016	*In vitro*/7 h Human neuroblastoma IMR 32 cells (cell no. TKG0207)	0–20 μM Eagle’s Minimum Essential Medium	Revealed gene expression profiles of Wasabi-derived ITCs in a neuronal cell model, IMR-32. 6-MSITC had the strongest regulation on gene expression among the three ITCs. 6-MSITC could stimulate Nrf2 mediated gene expressions through the stabilization of Nrf2 protein at post transcription. 6-MSITC exerted the neuroprotective effect by activating the Nrf2-mediated oxidative stress response pathway.
14.	[32]/2017	*In vitro*/9 h Human hepatoblastoma Hep2G cells (cell no. TKG0205)	10 mM Dulbecco’s Modified Eagle’s Medium	6-MSITC was found to be the most potent inducer of the Nrf2-dependent pathway, suggesting an important role of sulfinyl sulfur and carbon chain length in a liver cancer cell model. Furthermore, glutamate metabolism has also been shown to be regulated by 6-MSITC. 6-MSITC exerts a chemopreventive role against cancer through its primary antioxidant activity, activation of Nrf2, and subsequent induction of antioxidant proteins and metabolizing enzymes.
15.	[33]/2018	*In vitro*/48 h Human colorectal cancer cell lines HCT116 p53^+/+^	0 or 20 μM Dulbecco’s Modified Eagle’s Medium	6-MSITC inhibited cell proliferation and induces apoptosis in both HCT116 p53^+/+^ and HCT116 p53^−/−^ cells through a p53-independent mitochondrial dysfunction pathway. These results suggest that 6-MSITC may be a potential compound for the chemoprevention of colorectal cancer, even in the presence of a p53 mutation.
16.	[34]/2018	*In vivo*/29 days Male mice C57B1/6 (9 weeks old, 25–30g body weight at the beginning of the experiment)	5mg/kg every day Standard diet Free access	6-MSITC counteracted Aβ1-42 neurotoxicity in mice. These results highlight the interesting neuroprotective activity of 6-MSITC, which reduced apoptosis and neuroinflammation, restored physiological oxidative status, and positively influenced the Nrf2 pathway, resulting in significant behavioral improvement in our Alzheimer’s disease model. These data are promising, but further experimental studies are necessary to confirm the mechanism of action of 6-MSITC, assess its short- and long-term effects, and test its effectiveness in combination with other therapies.
17.	[35]/2019	*In vitro*/48 h two types of human colorectal cancer cells (HCT116 p53^+/+^ and p53^−/−^)	20 μM 6-MSITC Dulbecco’s Modified Eagle’s Medium	6-MSITC was shown to induce cell cycle arrest and apoptosis in both HCT116 p53^+/+^ and HCT116 p53^−/−^ cells, achieving the same effect. Molecular data showed that activation recruits ERK1/2 and not p53. It has been suggested that 6-MSITC-induced apoptotic cell death via the ELK1/CHOP/DR5 pathway via ERK1/2 is involved in molecular mechanisms.
18.	[36]/2020	*In vivo*/12 h + 48 h zebrafish larvae wild-type (AB strain) Nrf2-mutant (nfe2l2afh318)	6-MSITC (2.5, 5, 10 μM) E3+medium	The activities of 6-(methylsulfinyl)hexyl isothiocyanate were involved in the reduction of arsenite toxicity. The antioxidant activities were all Nrf2-dependent.
19.	[37]/2020	*In vivo*/7 days Human hepatocellular carcinoma HepG2 cells 20 male, 8-week-old *in vitro* C57BL/6J mice randomly assigned to four groups of five each (control and three doses of 6-MSITC administration groups)	0–40 mg/kg (10, 20–40 mg/kg) Dulbecco’s Modified Eagle’s Medium Normal diet	6-MSITC increased liver ALDH2 enzyme activity and protein expression by activating the Nrf2/ARE pathway and alleviated acetaldehyde-induced cytotoxicity. Therefore, 6-MSITC may protect hepatocytes against acetaldehyde-induced cytotoxicity. This study represents a potentially effective strategy for preventing the abnormal reaction induced by the ingestion of wasabi extract in the setting of alcohol consumption
20.	[38]/2020	*In vivo*/10 months Mice WT, AppNLGF, AppNLGF Keap1^FA/FA^, AppNLGF Keap1^FA/-^	0.4 mg/mL dissolved in water for 10 months 15 mg/kg intraperitoneally (to evaluate the expression of the Nrf2 target *Nqo1* gene)	Nrf2 induction improved brain antioxidant function and attenuates pathological neuroinflammation in the AppNLGF mouse model. Additionally, this study provides significant evidence supporting the concept that Nrf2 activation inhibits the onset and/or progression of Alzheimer’s disease (AD), indicating that the Keap1-Nrf2 system is a promising target for drug development for neurocognitive disorders, including AD.
21.	[39]/2022	*In vivo*, randomized placebo-controlled, double-blind/4 days Human, eight male	9 mg Standard diet	Consumption of 6-MSITC does not affect the concentration of calpain-1 in the blood and does not cause muscle damage or changes in inflammatory markers.
22.	[40]/2022	*In vivo*/12 weeks Human, fifteen patients (three males, twelve females, age 20–58 years)	orally administered wasabi extract (9.6 mg of 6-MSITC/day)	6-MSITC improved PS, frequency of headache and myalgia, neurocognitive symptoms such as brain fog, difficulty finding appropriate words, photophobia, and psychological vitality in patients with ME/CFS. In accordance with the effects on subjective symptoms, it also improved the PPT and scores on the TMT-A. Currently, treatment for neuro-cognitive dysfunction in ME/CFS patients is lacking.
23.	[41]/2022	*In vitro*/76 h TR146 cell line human oral epithelial cell line	1.5625–25 μM 50 μM—influence the viability of TR146 cells. Ham’s F12 medium	6-MSITC could suppress IL-6 and CXCL10 production in TNF-α-treated human oral epithelial cells (TR146 cells) by inhibiting the activation of STAT3 and NF-κB pathways.
24.	[42]/2022	*In vivo*/16 weeks 48 male Wistar rats (8 to 9 weeks old) weighing 330–340 g	5% wasabi powder Group 1: corn starch diet (C)—16 weeks normal drinking water Group 2: corn starch diet supplemented with 5% wasabi powder (CW) − 8 weeks (C) + 8 weeks (CW) normal drinking water Group 3: high-carbohydrate, high-fat diet (H)—16 weeks 25% fructose (*w*/*v*) in drinking water Group 4: high-carbohydrate, high-fat diet supplemented with 5% wasabi powder (HW) − 8 weeks (H) + 8 weeks (HW) 25% fructose (*w*/*v*) in drinking water	Tasmanian wasabi attenuated the changes in acute inflammation in the heart and lipid deposition in the liver and attenuated hypertension and obesity in diet-induced metabolic syndrome in Wistar rats.
25.	[43]/2022	*In vitro*/12 h, 24 h/48 h Human hepatocellular carcinoma HepG2 cells	0–20 M Dulbecco’s Modified Eagle’s Medium	6-MSITC significantly alleviated CFA-induced formation of thiobarbituric acid-reactive substances and fat accumulation. 6-MSITC enhanced phosphorylation of AMPKα, upregulated the expression of Nrf2, NQO1, heme oxygenase 1, FOXO1, and Siruin1, and downregulated the expression of PPARα. The AMPKα/Nrf2-mediated signaling pathways might be involved in the cytoprotective effects of Wasabi 6-MSITC against metabolic lipid stress.
26.	[44]/2023	*In vivo*/12 weeks Human 60 years and over 19 male 53 female	one 6-MSITC capsule that contained 100 mg wasabi extract powder containing 6-MSITC (0.8 mg)	Consumption of 0.8 mg of 6-MSITC for 12 weeks significantly improved memory functioning, including episodic and working memory, compared with the placebo group, but we did not observe significant improvements in other cognitive functions. This study is the first to demonstrate that 6-MSITC benefits memory function in healthy older adults.
27.	[45]/2023	*In vivo*/4 weeks Human, 20 healthy volunteers who were experiencing daily fatigue	powder containing 6-MSITC (4.8 mg/day of 6-MSITC)	6-MSITC did not improve fatigue after a mental task, but fatigue before the mental task, sleep, and mood were improved significantly. No changes were observed in autonomic nerve function, stress, or immune markers.
		*In vivo*, double-blind, placebo-controlled method/4 weeks Human overdose safety: 30 healthy volunteers	the extract powder (up to 16 mg/day of 6-MSITC for 4 weeks)	No changes in the parameters or side effects were observed, and the results showed that high doses of the extract powder containing 6-MSITC were safe.

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
