# Peer review of "Methylsulfinyl Hexyl Isothiocyanate (6-MSITC) from Wasabi Is a Promising Candidate for the Treatment of Cancer, Alzheimer’s Disease, and Obesity"

_nutrients, 2024, doi:10.3390/nu16152509_

Round 1

Reviewer 1 Report

Comments and Suggestions for Authors

The review effectively outlines the potential applications of 6-MSITC in treating various diseases, including cancer, metabolic syndrome, cardiovascular diseases, diabetes, obesity, and neurodegenerative diseases. The discussion on the neuroprotective properties and cognitive benefits is particularly compelling and well-supported by existing literature.

Although in vitro and in vivo animal studies have shown encouraging results, more clinical trials involving human participants are necessary to confirm the effectiveness and safety of 6-MSITC derived from Wasabi. Kindly provide one key aspect of the clinical and preclinical investigations of Wasabi.

Areas requiring improvement:

Expansion of Human Studies: When discussing human studies, it is important to highlight the necessity of conducting more comprehensive human trials in order to draw definitive results.

It would be beneficial to include a part in the discussion that focuses on the potential dosages and delivery mechanisms for 6-MSITC as a treatment.

Safety Considerations: It would be wise to provide a concise analysis of any established or possible adverse reactions associated with 6-MSITC.

Comparing 6-MSITC therapy with existing therapeutic choices for each disease application could enhance its potential role.

Future Directions: The text would be improved by including a more comprehensive analysis of the potential future paths for 6-MSITC research. Providing precise guidelines for the design and execution of human clinical trials would offer useful direction to researchers in the field.

Critical Assessment: Although the analysis is comprehensive, incorporating a more discerning evaluation of the constraints and difficulties in the existing research on 6-MSITC would enhance the depth of the discussion. Examining and acknowledging possible challenges and areas of limited understanding can offer a more detailed and sophisticated understanding of the molecule's potential for medical treatment.

In summary, this review study greatly enhances our comprehension of 6-MSITC and its potential applications in the field of medicine. With a few slight modifications, it possesses the potential to become an immensely significant asset for researchers and healthcare practitioners.

Author Response

Dear Reviewer,

We sincerely thank you for your valuable comments regarding our manuscript. Our responses can be found in the attached file.

Best regards,

Joanna Bartkowiak-Wieczorek

Reviewer 2 Report

Comments and Suggestions for Authors

The manuscript is not even submitted in the MDPI template. The English language is poor. The font is different throughout the manuscript. All the abbreviations should be explained under the figure.

I recommend a major revision of this poorly prepared manuscript before I can even judge the quality of the review itself.

Comments on the Quality of English Language

Native English speaker should proofread this manuscript

Author Response

(The authors gave the same response as above.)

Round 2

Reviewer 2 Report

Comments and Suggestions for Authors

I believe this version is now acceptable for publication

Author Response

Dear Reviewer,

I want to express our sincere gratitude for your positive review of our manuscript. Your feedback is greatly appreciated and has been very encouraging.

Thank you once again for your time and effort in reviewing our work.

Best regards,